# Ultrastructural Morphology and Descriptive Analysis of Cuticular Sensilla in Adult *Tomicus pilifer* (Coleoptera: Curculionidae)

**DOI:** 10.3390/insects16090890

**Published:** 2025-08-26

**Authors:** Longzheng Wang, Qi Wang, Yanan Luo, Shanchun Yan

**Affiliations:** 1Key Laboratory of Sustainable Forest Ecosystem Management, Ministry of Education, Northeast Forestry University, Harbin 150040, China; wanglzchina@163.com (L.W.); qiqi8325@163.com (Q.W.);; 2Forest Protection Research Institute of Heilongjiang Province, Heilongjiang Academy of Forestry, Harbin 150040, China

**Keywords:** sexual dimorphism, sensilla, ultrastructural, scanning electron microscopy

## Abstract

Insects heavily rely on chemical signals for host selection and mate finding, and sensilla, as the basic units of the sensory system, are distributed across various parts of the insect body, playing a crucial role in the reception and transmission of chemical signals. In this study, we utilized scanning electron microscopy (SEM) to investigate the types, numbers, sizes, and distributions of sensilla on the antennae, mouthparts, and legs of both male and female *Tomicus pilifer* adults. Additionally, by comparing previous studies on the *Tomicus* genus and other coleopteran insects, we discussed the potential functions of each sensillum type. These findings will provide a foundation for further research into insect–insect and plant–insect pheromone interactions.

## 1. Introduction

*Tomicus pilifer* is an important pest that threatens *Pinus koraiensis*, exhibiting a habit of transferring damage throughout its life cycle, which allows it to harm different parts of the tree. Adults lay eggs in the phloem, arranged neatly along the sides of the main galleries. After hatching, the larvae bore perpendicular to the main gallery, while the pupal stage remains concealed within the phloem of *Pinus koraiensis* until the adult emerges through an exit hole. The adult then migrates to the base of the new shoots of *Pinus koraiensis* for further nourishment [1,2,3]. This feeding behavior causes significant shoot drop within just one growing season, weakening the tree significantly. Without timely intervention, it can also attract other pest species, exacerbating its damage. In addition to harming *Pinus koraiensis*, *T. pilifer* also exhibits feeding behavior on *Pinus armandii*, *Pinus tabulaeformis*, and *Pinus yunnanensis* [1,2]. This species is recorded to be distributed only in Asia (Russia and China). Currently, *T. pilifer* causes damage to *Pinus koraiensis* forests in various regions of northeast China, severely affecting local forests’ health and economic development [3]. However, control methods for *T. pilifer* still largely rely on chemical pesticides, with limited research and promotion of green, low-pollution chemicals. As a result, pesticide choices for *T. pilifer* are still limited to 40% oxydemeton–methyl emulsion, 80% dichlorvos emulsion, and 80% phosalone emulsion, which not only cause environmental pollution but also have adverse effects on human health [4]. Therefore, there is an urgent need for effective and safe control measures to manage *T. pilifer*. However, there have been no reports on the sensillum morphology and distribution of *T. pilifer*, and research on its olfactory recognition mechanisms is lacking. This greatly limits the development and application of pest control technologies based on insect sensory behavior.

Research on the morphology and structure of insect sensilla is fundamental to understanding their olfactory recognition mechanisms and behavior [5,6,7]. Insects use highly specialized sensilla to detect physical and chemical stimuli from the external environment [8,9], enabling them to identify hosts. These sensilla are distributed across the insect’s body, primarily concentrated on the antennae, mouthparts, and legs, acting as the “windows” through which insects interact with their surroundings [5,10,11]. The antennae serve as the main sensory organs in insects, with various types of sensilla present on their surface [12,13,14,15]. Based on their physiological functions, sensilla can be classified into mechanoreceptors, chemoreceptors, and thermohygroreceptors [16,17,18]. Some sensilla have micropores on their surface, particularly chemoreceptors [19,20,21], through which odor molecules pass into the sensillar lymph and are transported by odorant-binding proteins (OBPs) to the vicinity of the odor receptors on the dendritic membranes of sensory neurons. After a series of reactions, chemical signals are converted into electrophysiological signals and transmitted to the central nervous system [22,23,24], thus regulating behaviors such as feeding, host location, oviposition, predator avoidance, and mate seeking [25,26,27,28]. The mouthparts are the feeding organs of insects and also contain various sensillar structures, with sensilla on the maxillary palps and labial palps playing a crucial role in host and food source recognition [29,30,31]. Compared to the antennae and mouthparts, research on the sensilla on the legs of insects is relatively limited. However, some studies suggest that various sensilla on the legs contribute to taste, olfaction, and thermohygroreception [31,32].

This study used scanning electron microscopy (SEM) to examine the morphology, types, and distribution of sensilla on the antennae, mouthparts, and legs of both male and female *Tomicus pilifer* adults. The abundance and distribution of sensilla were compared and analyzed between the sexes. The findings provide a foundation for further exploration of the olfactory mechanisms underlying host selection, feeding, and mate-seeking behaviors in *T. pilifer*. Additionally, this study offers theoretical guidance for the development of novel control strategies for *T. pilifer*.

## 2. Materials and Methods

### 2.1. Insect

In early June 2024, infested *Pinus koraiensis* logs were collected from the Jiulong Village Forestry Farm, Boli County, Harbin City, Heilongjiang Province (130°26′50″ E, 45°43′35″ N). The logs were sealed with wax at both ends and incubated indoors. Freshly emerged adults were collected at regular intervals. Healthy male and female adults that had eclosed were placed separately into small rearing bottles and stored at 4 °C for further use.

### 2.2. Molecular Identification

Molecular identification was performed using the DP304-2 kit from TianGen Biotech (Beijing) Co., Ltd. (Beijing, China), following the manufacturer’s instructions. The D2 region of the 28S subunit of the rDNA from the beetle samples was amplified using the following primers: 28Sscol1: 5′-AACGAAAGGTCGAAGGAAG-3′; and D3R: 5′-TAGTTCACCATCTTTCGGGTC-3′. The PCR conditions were as follows: initial denaturation at 94 °C for 5 min, followed by 35 cycles of 94 °C for 30 s (denaturation), 50 °C for 40 s (annealing), and 72 °C for 1 min (extension), with a final extension at 72 °C for 10 min. The PCR products were directly sequenced, and sequencing was performed by Beijing Ruibo Xingke Biotechnology Co., Ltd. (Beijing, China). The obtained sequences were assembled using DNAMAN v7.0 (Lynnon Biosoft, Canada) Homologous sequence searches were performed using blast from GenBank. The selected species for alignment and their corresponding GenBank accession numbers are listed in Appendix A. Phylogenetic analysis was conducted using the Maximum Likelihood (ML) method in MEGA v11.0 (Appendix A), confirming the identification of the insect as *Tomicus pilifer*.

### 2.3. Scanning Electron Microscopy

The antennae, mouthparts, forelegs, midlegs, and hindlegs of both male and female *Tomicus pilifer* adults were dissected using fine forceps and an anatomical needle under a stereomicroscope (Olympus, SZX16, Tokyo, Japan). Each body part was placed into an ultrasonic cleaner and washed with physiological saline for 30 s, repeating the process three times. Subsequently, the samples were subjected to a gradient dehydration process using different concentrations of ethanol (30%, 50%, 60%, 70%, 80%, 90%, and 100%), with each stage lasting 15 min. After natural drying at room temperature, the samples were mounted on specimen holders using double-sided tape and sputter-coated with gold. Scanning observations were conducted using a scanning electron microscope (Apreo C, Thermo Fisher Scientific, Waltham, MA, USA) within a voltage range of 3–10 kV.

### 2.4. Terminology and Statistical Analysis

The sensillum classification and identification for *Tomicus pilifer* were based on the nomenclature standards established by Schneider [16], Brooks [33], and Keil [34], and were compared and analyzed with previous studies on sensilla in other *Tomicus* species. Image contrast and brightness were adjusted using Adobe Photoshop 2022 (Adobe Systems, San Jose, CA, USA), and extraneous background impurities were removed. The length of each sensillum type was measured using ImageJ v1.53 (NIH, Bethesda, MD, USA), with at least six specimens of the same sensillum type selected for each measurement and analysis. All statistical analyses were performed using SPSS Statistics 27.0 (IBM, Armonk, NY, USA).

## 3. Results

### 3.1. Antennal Morphology and Sensilla Types of T. pilifer

The antennae of both male and female *T. pilifer* adults are club-shaped, with no evident sexual dimorphism in their morphological structure (Figure 1). The male antennae measure 612.7 ± 34.7 μm in length, while the female antennae are 649 ± 39.5 μm long. Both sexes’ antennae are divided into three segments: the scape, funicle, and capitate (Figure 1A). The scape is a single, thickened segment with a club-like shape, measuring approximately 214.7 ± 16.3 μm in length and about 73.4 ± 9.6 μm in width at its widest point. The funicle consists of six segments, labeled F1–F6 (Figure 1B), with F1 being bead-like and F2–F6 forming a tubular structure. From F2 onwards, each segment of the funicle gradually increases in thickness (Figure 1B). The clubbed region is rod-like, divided into four segments (C1–C4), with the segments being straight between each node. The club measures approximately 194.3 ± 13.8 μm in length and about 156.4 ± 19.8 μm in width at its widest point (Figure 1B).

Each segment of the antennae bears different types and quantities of sensilla. Specifically, the scape and funicle have fewer and more sparsely distributed sensilla, with the surface exhibiting distinct polygonal or corrugated patterns. In contrast, the clubbed region has a higher density of sensilla, arranged in a regular, circumferential pattern between the segments.

A total of six types of sensilla, including eleven subtypes, were identified on the antennae of *Tomicus pilifer*. These included sensilla trichoidea I and II (ST I and II), sensilla zigzag I and II (SZ I and II), sensilla coeloconica (Sco), sensilla chaetica (Sch), Böhm bristles (Bb), and sensilla basiconica I, II, III, and IV (Sb I, II, III, and IV). Notably, sensilla basiconica III were found exclusively on male antennae. Additionally, on segment F2 of the funicle, the male antennae were observed to be smooth, with no sensilla present; in contrast, sensilla chaetica were observed at the corresponding location on the female antennae (Table 1).

#### 3.1.1. Sensilla Trichoidea (ST)

Sensilla trichoidea are exclusively distributed in the band-like depressions of the clubbed region of both male and female antennae (Figure 2A). These sensilla are classified into two types, s. trichoidea I and s. trichoidea II, based on differences in size, curvature, and exine. S. trichoidea I is primarily found on segments C1–C3 of the club, arranged in a relatively neat, whorled pattern, and becomes more densely scattered from segment C4 onwards, with a few scattered across the intersegmental areas of the capitate (Figure 2). S. trichoidea I measures 15.6 ± 1.7 μm in length, with a basal diameter of 1.4 ± 0.3 μm. It is oriented perpendicular to the base, with a tapered distal end, and the surface exhibits a bundled texture (Figure 2B). S. trichoidea II is the longest of all sensilla types, distributed both along the sensillar band and at the apex of the club. It measures 34.3 ± 3.2 μm in length and has a basal diameter of 2.1 ± 0.4 μm. The sensillum is slightly curved, and its abundance is significantly lower than that of s. trichoidea I. No significant difference in the abundance of s. trichoidea was observed between male and female antennae (Figure 2C; Table 2).

#### 3.1.2. Sensilla Zigzag (SZ)

Sensilla zigzag are the most widely distributed sensilla, found at the base of the scape, funicle, and capitate (Table 1). These sensilla are hair-like, sparsely distributed, and may be upright or bent. The surface of these sensilla is characterized by saw-like teeth, and they are situated in relatively deep basal pits. Based on their length, shape, and degree of curvature, they are classified into three subtypes. S. zigzag I are distributed at the base of the scape and capitate (Figure 2A). Typically, they have two to nine teeth on one side, while the other side is smooth. The length of s. zigzag I is 23.1 ± 12.7 μm, with a basal diameter of 2.7 ± 0.4 μm, and the contact angle is 67.4 ± 8.9° (Figure 3A). S. zigzag II is found only on the funicle, and compared to s. zigzag I, it has shorter teeth, with two pairs of teeth opposite each other starting from the second tooth at the distal end. S. zigzag II measures 15.3 ± 1.9 μm in length, with a basal width of 1.3 ± 0.4 μm, and a contact angle of 37.2 ± 11.4° (Figure 3B). Although the abundance of s. zigzag is much lower than that of s. trichoidea, their distribution range on the antennae is broader (Table 1). The abundance of s. zigzag I exceeds that of s. zigzag II (Table 2).

#### 3.1.3. Sensilla Coeloconica (Sco)

Sensilla coeloconica are flower bud-shaped, measuring 8.6 ± 0.7 μm in length, with a basal width of 2.7 ± 0.2 μm. They are exclusively distributed on the capitate of the antennae and are often surrounded by sensilla trichoidea and sensilla basiconica. The number of these sensilla is relatively low. Each sensillum exhibits a distinct morphological difference between its upper and lower parts: the base is conical and smooth, while the upper part resembles a flower bud, encased by filamentous sensillar walls. The distal end gradually narrows and converges, resembling an unopened onion bud, with clear longitudinal striations visible on its surface (Figure 4A; Table 2).

#### 3.1.4. Sensilla Chaetica (Sch)

Sensilla chaetica arise from a relatively thick basal groove, with a noticeable gap between each sensillum and its basal groove. The entire surface of the sensillum is covered with spiral longitudinal grooves that taper sharply toward the distal end, which is pointed. S. chaetica I measures 7.6 ± 0.7 μm in length, with a basal width of 2.9 ± 0.2 μm. This type of sensillum is primarily located on the clubbed region of the antennae, interspersed with sensilla trichoidea, but its abundance is significantly lower than that of sensilla trichoidea (Figure 4B; Table 2).

#### 3.1.5. Böhm Bristles (Bb)

Böhm bristles are upright, conical in shape, with a thicker base and a rounded, blunt distal end. They measure approximately 2.4 ± 0.3 μm in length, with a basal diameter of about 1.1 ± 0.1 μm. This sensillum is the shortest among all sensilla, and it is predominantly concentrated in the depressions at the base of the scape (Figure 4C; Table 1 and Table 2).

#### 3.1.6. Sensilla Basiconica (Sb)

Sensilla basiconica are cone-shaped and represent the most abundant type of sensilla on the antennae, found exclusively in the sensillar band of the intersegmental regions of the capitate. The surface of these sensilla is rough, and based on their shape and surface sculpture, they are classified into four subtypes: s. basiconica I, s. basiconica II, s. basiconica III, and s. basiconica IV. S. basiconica I is the most abundant sensillum within this type, with a surface covered in numerous dot-like micropores. The apex is sharp, measuring approximately 11.2 ± 0.9 μm in length, and the basal width is 2.3 ± 0.2 μm (Figure 5A; Table 2). S. basiconica II is similar in size to s. basiconica I, but it exhibits a noticeable constriction near the distal end and features a bundled, crossed surface texture (Figure 5B). S. basiconica I and S. basiconica II are interspersed and distributed along the C1-C3 sensillar bands of the capitate, where they are enveloped by sensilla trichoidea. S. basiconica III and s. basiconica IV are both located in the sensillar band surrounding the apex of the capitate. S. basiconica III is found exclusively on the male antennae, with no occurrence in females. This sensillum is bent inward, with a rough surface and numerous micropores, and it forms a ring-like arrangement at the apex of the capitate (Figure 5C). S. basiconica IV is thumb-shaped, with a cap-like distal end, and is sparsely distributed in the sensillar band at the end of the capitate. It measures approximately 5.7 ± 0.6 μm in length, with a basal width of 1.6 ± 0.2 μm, and is less numerous than the other subtypes (Figure 5D).

### 3.2. The Mouthpart Structure and Sensilla Types of T. pilifer

The mouthparts of *Tomicus pilifer* adults are typical chewing-type mouthparts, consisting of a labrum, two mandibles, two maxillae, and one labium (Figure 6A). The labrum is highly reduced and has a simple, plate-like structure. The mandibles are positioned beneath the labrum and possess a highly sclerotized, hard conical shape. The inner surface of the conical tip is serrated, serving the functions of grinding and cutting food (Figure 6B). The maxillae consist of five parts: the basal segment, the stem, the outer maxillary lobe, the inner maxillary lobe, and the maxillary palps. The maxillary palps are divided into three segments, with various types of sensilla distributed on each segment (Figure 6C,D). The labium is situated between the two mandibles, with two robust labial palps at the distal end. The labial palps are divided into three segments, and the types of sensilla on their surface are similar to those found on the maxillary palps (Figure 6E,F).

A total of seven sensillum types, comprising thirteen subtypes, were observed on the mouthparts of both male and female adults. These included sensilla basiconica I and II (Sb I, II), sensilla twig basiconica I, II, and III (Stb I, II, and III), sensilla coeloconica (Sco), sensilla trichoidea I and II (ST I, II), sensilla chaetica (Sch), sensilla zigzag I, II, and III (SZ I, II, and III), and sensilla digitiformia (Sdi).

#### 3.2.1. Sensilla Basiconica (SB)

Sensilla basiconica are conical in shape and upright, with two subtypes: s. basiconica I and s. basiconica II. S. basiconica I has a smooth surface with a blunt distal end, featuring micropores. It measures approximately 5.2 ± 1.2 μm in length and 0.8 ± 0.2 μm in basal width. These sensilla are located exclusively at the distal ends of the maxillary and labial palps, where they are the largest sensilla in the distal sensillar field (Figure 7A,B,D). S. basiconica II is significantly more abundant and longer than s. basiconica I (Table 3). It is cone-shaped, with a length of approximately 28.7 ± 5.3 μm and a basal width of 5.1 ± 1.2 μm. These sensilla are clustered and concentrated at the ligula of the inside of the labium (Figure 6F and Figure 7C).

#### 3.2.2. Sensilla Twig Basiconica (Stb)

Sensilla twig basiconica are the second-largest sensilla in the distal sensillar field, densely distributed at the tips of the mandibular and labial palps. These sensilla are cone-shaped, with considerable variation in the shape of the conical tip, and can be divided into three subtypes: s. twig basiconica I, s. twig basiconica II, and s. twig basiconica III (Figure 7A,B). The apex of s. twig basiconica I features orderly arranged digitiform projections in the cuticular layer, which converge towards the center. The length of s. twig basiconica I is approximately 2.4 ± 0.2 μm, with a basal width of about 1.2 ± 0.1 μm (Figure 7D). S. twig basiconica II is similar in appearance to s. twig basiconica I, but the distal end has a concave pore resembling a volcanic crater. The length of s. twig basiconica II is approximately 3.1 ± 0.2 μm, with a basal width of about 0.9 ± 0.1 μm (Figure 7E). S. twig basiconica III is the smallest sensillum in the distal sensillar field, featuring a double-layered structure with an outer ring-like arrangement. S. twig basiconica III is located at the edge of the apical tip of the palp and arranged in a ring. The length of s. twig basiconica III is approximately 1.4 ± 0.2 μm, with a basal width of about 1.2 ± 0.1 μm (Figure 7F).

#### 3.2.3. Sensilla Coeloconica (Sco)

Sensilla coeloconica (Sco) are conical protrusions, with multiple sensilla located in a wide, circular groove. The conical structure is small, measuring approximately 1.7 ± 0.2 μm in length and 0.8 ± 0.1 μm in width. These sensilla are found only on the lateral sides at the distal end of the maxillary palps, with a typical count of two to three sensilla (Figure 7B).

#### 3.2.4. Sensilla Trichoidea (ST)

Sensilla trichoidea are hair-like structures, and are classified into two subtypes based on their length: s. trichoidea I and s. trichoidea II. S. trichoidea I has a smooth surface and is upright or slightly curved at the tip, with a length of approximately 87.1 ± 21.2 μm and a base width of around 5.1 ± 1.2 μm. These sensilla are distributed on the outer side of the maxilla and outer maxillary lobe (Figure 6C and Figure 8A). S. trichoidea II is similar in shape to s. trichoidea I but differs significantly in length. S. trichoidea II is much shorter than s. trichoidea I, with a length of approximately 14.2 ± 2.6 μm and a base width of about 2.3 ± 0.7 μm. These sensilla are located on the ventral surface of the inner maxillary lobe and the side of the stem segment (Figure 6C and Figure 8B).

#### 3.2.5. Sensilla Chaetica (Sch)

Sensilla chaetica are rigid and spine-like in appearance, with a smooth surface. They are located in a relatively wide, circular groove, measuring approximately 14.2 ± 2.7 μm in length and 1.7 ± 0.4 μm in basal width. These sensilla are distributed on the lateral sides of the maxilla, outer maxillary lobes, and labrum (Figure 6C,F and Figure 9).

#### 3.2.6. Sensilla Zigzag (SZ)

Sensilla zigzag resemble hair and are adorned with serrations. Based on the shape and number of serrations, they are classified into three subtypes: s. zigzag I, s. zigzag II, and s. zigzag III. S. zigzag I is located in a deep, circular groove and begins to develop serrations approximately one-third from the base. Each serration is sharp at the tip, with one serration per segment. The length of this subtype is approximately 34.8 ± 2.1 μm, with a basal width of about 1.9 ± 0.3 μm, and it is distributed on the stem segment, outer maxillary lobes, and labium (Figure 6E and Figure 10A). S. zigzag II starts to form serrations approximately two-thirds from the base and curves into an “S” shape. It is found in a cupped groove, with serrations clustered and rounded at the tips. The length of s. zigzag II is approximately 107.3 ± 19.6 μm, with a basal width of about 5.9 ± 0.7 μm, and it is exclusively distributed on the outer maxillary lobes (Figure 6C and Figure 10B). S. zigzag III differs morphologically from both s. zigzag I and s. zigzag II. It develops three branches from the base of the sensor, with the main shaft standing upright and serrated. This subtype is less abundant in number than s. zigzag I and s. zigzag II, and it is found only on the lateral sides of the labium (Figure 6E and Figure 10C).

#### 3.2.7. Sensilla Digitiformia (Sdi)

Sensilla digitiformia are located exclusively on the lateral surface of the third subsegment of the maxillary palp (Figure 11A), opposite to the distribution of sensilla coeloconica (Figure 11B). Sensilla digitiformia are plate-like or finger-shaped, lying flat within a long elliptical epidermal groove, typically arranged in five parallel structures. Their length is approximately 13.7 ± 0.8 μm, with a width of about 1.3 ± 0.2 μm (Figure 11C).

### 3.3. Sensilla Types and Morphology on Legs

In both male and female adults of *Tomicus pilifer*, three types of sensilla with six subtypes were observed on the forelegs, midlegs, and hindlegs, including sensilla zigzag I and II (SZ I, II), sensilla trichoidea (ST), and sensilla chaetica I and II (Sch I, II). No significant differences in the types of sensilla were found across the forelegs, midlegs, and hindlegs. Therefore, the forelegs of *Tomicus pilifer* are used as an example to illustrate the structural features and types of sensilla (Figure 12). The number of each type of sensilla on the forelegs, midlegs, and hindlegs of male and female *T. pilifer* adults is shown in Table 4.

Each leg of *T. pilifer* is composed of three segments: the femur, tibia, and tarsus (Figure 12A). The distribution of sensilla on the inner and outer surfaces of the femur shows marked differences, with the outer surface densely covered with sensilla (Figure 12A,B). Additionally, at the junction between the tibia and tarsus, the outer surface exhibits lateral teeth, while the inner surface contains sensilla chaetica (Figure 12B). The tarsus consists of five segments, with segments 1–3 densely populated with sensilla. The fourth segment is small, embedded within the third segment, and lacks any sensilla. The fifth segment is longer, gradually widening from the base to the tip, with a few sensilla present, and ends in a large claw (Figure 12C,D).

#### 3.3.1. Sensilla Zigzag (SZ)

Sensilla zigzag are the most abundant type of sensilla on the legs of *Tomicus pilifer* adults (Table 4). They are distributed across the femur, tibia, tarsus, and fore-tarsus, with two subtypes: s. zigzag I and s. zigzag II. S. zigzag I is primarily found on the tibia and tarsus, where it is located in a cup-shaped groove. The sensillum is upright and serrated in appearance, with one to eight teeth on one side, while the other side is smooth. The length of s. zigzag I is approximately 42.3 ± 5.6 μm, with a base width of about 2.9 ± 0.3 μm (Figure 13A). S. zigzag II is mainly distributed on the outer surface of the femur, embedded in a scale-like groove. One side near the femur is smooth, while the opposite side bears irregular, interlaced serrations. The number of s. zigzag II is significantly lower than that of s. zigzag I, with a length of approximately 38.7 ± 4.3 μm and a base width of about 3.4 ± 0.7 μm (Figure 13B).

#### 3.3.2. Sensilla Trichoidea (ST)

Sensilla trichoidea are most abundant on the tarsus, concentrated at positions aligned with the direction of the fore-tarsal claw (Figure 12C), with sensilla zigzag sparsely distributed on their dorsal surface. Based on external morphology, they are classified into two types: s. trichoidea I and s. trichoidea II. S. trichoidea I is found on the first and second segments of the tarsus, is hair-like, and has a smooth surface. It gradually tapers from the base to a sharp tip, with a length of approximately 37.6 ± 3.9 μm and a basal width of about 2.1 ± 0.5 μm (Figure 14A,B). S. trichoidea II flattens and widens along its length, with a bundled spiral texture on the surface, resembling a bean sprout. It measures approximately 46.8 ± 5.6 μm in length and 18.2 ± 0.3 μm in basal width, and it is exclusively distributed on the third segment of the tarsus, being less abundant than s. trichoidea I (Figure 14; Table 4).

#### 3.3.3. Sensilla Chaetica (Sch)

Sensilla chaetica are the second-largest type of sensilla on the legs, classified into two subtypes: s. chaetica I and s. chaetica II. S. chaetica I is primarily distributed on the tibia and tarsus, with a smooth surface, growing upright towards the tarsus. It gradually tapers from the base to a sharp tip, measuring approximately 35.4 ± 4.5 μm in length and 3.7 ± 0.6 μm in basal width (Figure 15A). S. chaetica II is the most robust type of sensillum, exclusively located at the junction between the tibia and tarsus. A row of these sensilla encircles the edge of the tibia, with a surface featuring longitudinal striations and a rounded tip. The length of s. chaetica II is approximately 68.5 ± 3.7 μm, with a basal width of about 8.6 ± 1.2 μm (Figure 15B,C).

## 4. Discussion

The antennae of *Tomicus pilifer* are club-shaped, consisting of three parts: the scape, funicle, and capitate. No obvious sexual dimorphism is observed in their morphological structure. Additionally, a total of six types of sensilla, including twelve subtypes, were identified on the antennae of *T. pilifer*. These include sensilla trichoidea I and II, sensilla zigzag I and II, sensilla coeloconica, sensilla chaetica, Böhm bristles, and sensilla basiconica I, II, III, and IV. Compared to previously reported sensilla types on the antennae of *Tomicus* species [35], we identified three new subtypes of sensilla basiconica (II, III, and IV) on *T. pilifer* antennae. Notably, sensilla basiconica III were observed only on the male antennae. At segment F2 of the funicle, we found the male antennae to be smooth, without any sensilla, while the same position on the female antennae exhibited sensilla chaetica. These results indicate the presence of sexual dimorphism in the types and distribution of sensilla on *T. pilifer* antennae. Furthermore, no significant differences in the size and number of each sensillum type were observed between the sexes, suggesting a high degree of functional similarity between the sensilla of males and females. Similar results have been reported in studies of other coleopteran insects, which may be related to specific environmental conditions [11,16,36,37].

Sensilla trichoidea (ST) are primarily concentrated in the sensillar band of the capitate region of the antennae, with few found on the funicle and scape. This distribution pattern is similar to those reported by Wang [35] and Chen [38]. ST I is located at the outermost edge of the sensillar band in the capitate, with a surface featuring vertical striations and a robust structure. This is highly similar to the ST I observed on the antennae of *Tomicus yunnanensis* and *Tomicus piniperda*, where they are commonly considered to be a mechanoreceptor and chemoreceptor [39]. Additionally, they may also play a protective role for the sensillar field, including sensilla basiconica and sensilla coeloconica [35,40]. ST II is the longest sensillum on the antennae, mainly distributed at the distal end of the capitate. It has a surface with interlaced vertical striations, resembling the ST observed in *Megabruchidius dorsalis* [41], *Ips typographus* [42], and *Bruchidius coreanus* [43]. This suggests that ST II may play a role in collecting and detecting sex pheromones or aggregation pheromones [37,44]. Sensilla zigzag are the most widely distributed type of sensilla, found on the scape, funicle, and capitate of the antennae. Similar findings have been reported in many *Curculionidae* species [42,45,46]. It is worth mentioning that due to the significant morphological differences and variety of subtypes observed in sensilla zigzag, we did not classify them as a subtype of sensilla chaetica, as was suggested by Shi [42] and Shewale [45], but instead treated them as a separate sensilla type, as suggested by Chen [46] and Lu [47]. Zacharuk [48] proposed that these unique serrated sensilla often function as both contact chemoreceptors and mechanoreceptors. Sensilla coeloconica have a distinctive flower bud-like shape, with longitudinal grooves on the walls, and have been reported in many *Curculionidae* species [38,45,49]. Previous studies have shown that Sco in moths play an olfactory role [50], and Hallberg [51] discovered that the Sco on the antennae of *Ips typographus* have a dual-wall structure, which is similar to that of certain chemoreceptors [52]. Therefore, it is inferred that Sco may function in olfaction as well as thermohygroreception. Sensilla chaetica (Sch) and sensilla zigzag are introduced separately. Therefore, in this study, only one subtype of Sch was observed. Sch was found exclusively at the distal end of the funicle and the capitate, characterized by deep longitudinal striations. Some studies suggest that Sch is the most likely sensillum to function as both a mechanoreceptor and a chemoreceptor [6,53]. Böhm bristles (Bb), located on the scape, are the shortest type of sensilla on the antennae. These sensilla have a smooth surface and are arranged like small spines in clusters around the intersegmental regions of the scape and pedicel. In this study, the distribution and morphological characteristics of Bb were similar to those observed in *Curculionidae* insects. Wang [35] observed, through transmission electron microscopy, that these sensilla lack wall pores, suggesting they do not possess olfactory functions. Krishnan [54] hypothesized that Bb are mechanoreceptors that sense gravity, primarily functioning to detect mechanical impacts and buffer the effects of gravity. Sensilla basiconica represent the most abundant type of sensillum on the antennae, with the highest number of subtypes. These sensilla are densely distributed in the sensillar band of the intersegmental region of the club. The presence of numerous micropores and branched dendrites suggests that Sb may function as olfactory receptors, likely involved in odor recognition and host location [36,55,56]. Interestingly, we observed the unique Sb III only on male antennae. Our field observations revealed that during the breeding and mating period after overwintering, females are the first to bore entry holes into the host galleries for nutritional supplementation, followed by males, who enter through the female’s entry hole to feed and complete mating (Appendix A). We hypothesize that Sb III may play a role in collecting and detecting the sex pheromones released by females. Further studies are needed to elucidate its specific function.

We also observed two types of sensilla on the mouthparts of *Tomicus pilifer* that were strikingly similar to those found on the antennae: ST and Sch. These sensilla are widely distributed on the mouthparts and serve a mechanoreceptive function [41,57]. Additionally, we observed sensilla basiconica I and II; sensilla twig basiconica I, II, and III; sensilla coeloconica; and sensilla digitiformia at the distal ends of the maxillary and labial palps. The types, numbers, and distribution patterns of these sensilla are consistent with those previously reported in *Tomicus* species [58]. This suggests that the types, numbers, and distributions of mouthpart sensilla are highly similar within different species of *Tomicus*, and similar findings have also been reported in studies of *Ips* species’ mouthpart sensilla [42,43,51]. In terms of sensillum function, Cui [58] found that the cross-section of Sb exhibits a single-wall structure, with 2–3 dendritic nerves entering the lymph cavity. Therefore, it is inferred that an Sb located on the mouthparts is more likely a gustatory receptor, sensing the chemicals present in food during ingestion. Sensilla twig basiconica are the most abundant sensilla at the distal ends of the maxillary and labial palps. These sensilla may simultaneously function in mechanoreception and gustation [58,59]. Sensilla coeloconica and sensilla digitiformia are both located on the third segment of the maxillary palps, where they are arranged oppositely. Their distribution on the third segment of the labial palps is less frequent. Cui [58] observed branched tubular crystals in the cross-section of Sco and inferred that it may function as a mechanoreceptor [60]. Some studies suggest that Sdi is a poreless sensillum [51,61]. Based on the observation of numerous dendrites innervating the lymph cavity in the cross-section of Sdi in the mouthparts of *Tomicus*, it is inferred that Sdi may play a role in detecting vibrational signals for intraspecific communication [58,62].

Compared to the antennae and mouthparts of beetles, studies on the ultrastructure of sensilla on the legs are scarce. In *Tomicus pilifer*, we identified three types of sensilla with five subtypes on the legs, including sensilla zigzag I and II, sensilla trichoidea I and II, and sensilla chaetica. Among these, sensilla zigzag and sensilla chaetica are almost entirely distributed on the femur and tibia. These types of sensilla are also common on the antennae and mouthparts and are often considered to primarily function as mechanoreceptors [32,63]. Sensilla trichoidea (ST) are densely distributed on one side of the tarsus, the side that is in direct contact with the ground when the beetle is crawling. As they come into direct contact with surfaces, sensilla located on the tarsus are commonly believed to be involved in sensing tactile, gustatory, and mechanical stimuli, as well as in gravity [63,64,65].

## 5. Conclusions

In this study, we investigated the types, numbers, sizes, and distribution patterns of sensilla on the antennae, mouthparts, and legs of *Tomicus pilifer*. Notably, we identified three new subtypes of sensilla basiconica, sensilla basiconica II, III, and IV, on the antennae of *T. pilifer*, with sensilla basiconica III observed exclusively on male antennae. At segment F2 of the funicle, we found the male antennae to be smooth, with no sensilla present, while the same location on the female antennae exhibited sensilla chaetica. These findings indicate sexual dimorphism in the types and distribution of sensilla in *T. pilifer*. Our study further expands the research on the ultrastructure of sensilla in *Tomicus* beetles and aims to establish a standardized nomenclature for sensilla in this genus in the future. The findings of this study also lay the foundation for future research on the olfactory, feeding, and mating behaviors, as well as the electrophysiological aspects, of this destructive forest pest. Additionally, while speculating on the functions of sensilla in *Tomicus pilifer*, it was observed that the same type of sensillum may have different functions depending on its location. For example, in this study, the sensilla trichoidea on the antennae and mouthparts primarily function in mechanoreception, whereas those located on the pedicel may have a dual function, serving both mechanoreception and gustation. Therefore, further investigations using techniques such as antenna potential recordings, single-cell recordings, transmission electron microscopy, and molecular biology are necessary to deepen our understanding of the roles and functions of these sensilla in insect behavior.

## Figures and Tables

**Figure 1 insects-16-00890-f001:**
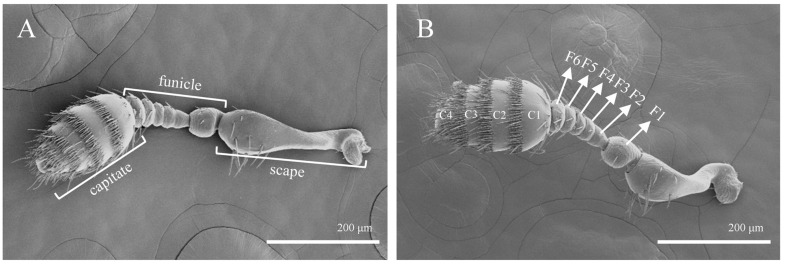
Morphology of *T. pilifer* antennae. (**A**) Male; (**B**) female; F1–F6: the 1st to 6th subsections of the funicle; C1–C4: the 1st to 4th subsections of the capitate.

**Figure 2 insects-16-00890-f002:**
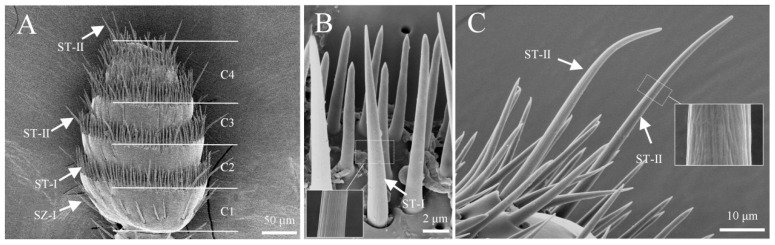
Morphology of sensilla trichoidea. (**A**) Sensillum distribution status of the capitate; (**B**) s. trichoidea I on the antennae; (**C**) s. trichoidea II on the antennae.

**Figure 3 insects-16-00890-f003:**
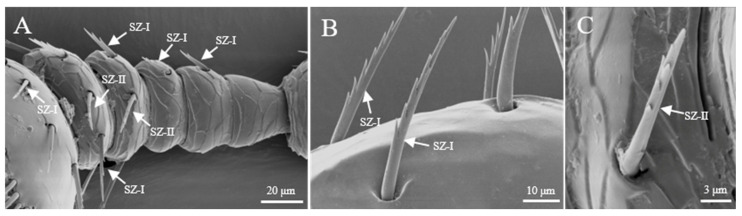
Morphology of sensilla zigzag on the antennae. (**A**) Distribution of sensilla zigzag on the funicle; (**B**) s. zigzag I on the antennae; (**C**) s. zigzag II on the antennae.

**Figure 4 insects-16-00890-f004:**
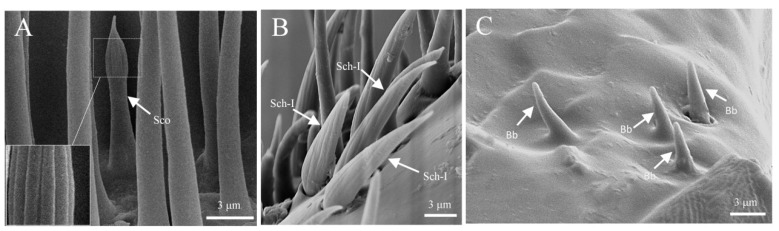
The ultrastructure of several sensilla. (**A**) S. coeloconica on the antennae; (**B**) s. chaetica on the antennae; (**C**) Böhm bristles on the antennae.

**Figure 5 insects-16-00890-f005:**
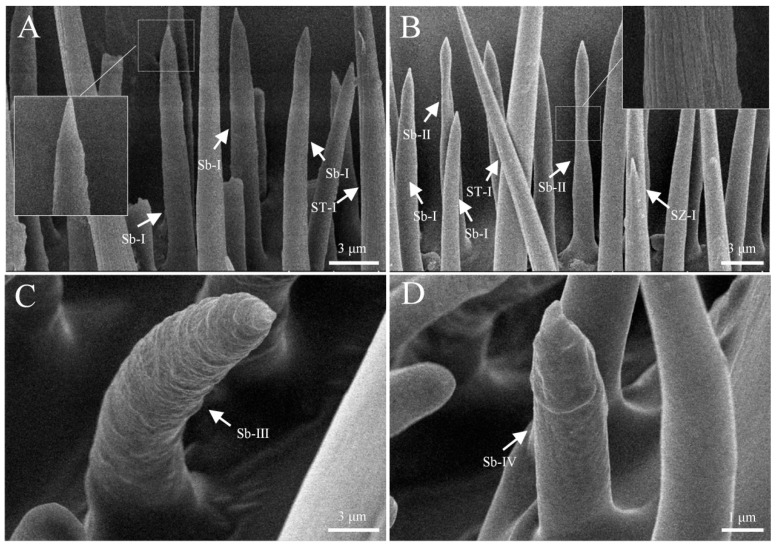
The ultrastructure of Sensilla basiconica. (**A**) S. basiconica I on the antennae; (**B**) s. basiconica II on the antennae; (**C**) s. basiconica III on the antennae; (**D**) s. basiconica IV on the antennae.

**Figure 6 insects-16-00890-f006:**
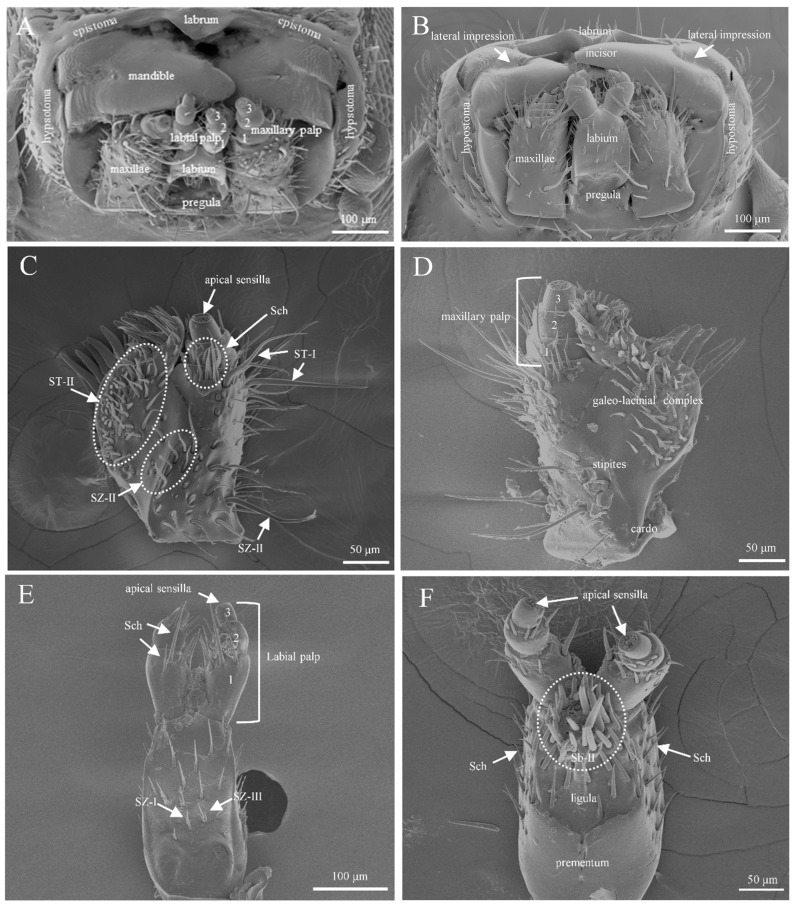
General morphology and structure of the mouthparts of *T. pilifer*. (**A**) The mouthparts of male insects; (**B**) the mouthparts of female insects; (**C**) the maxillae in ventral view; (**D**) the maxillae in dorsal view, 1–3: sections 1 to 3 of the maxillary palp; (**E**) the labium from external view, 1–3: sections 1 to 3 of the labial palp; (**F**) the labium from internal view.

**Figure 7 insects-16-00890-f007:**
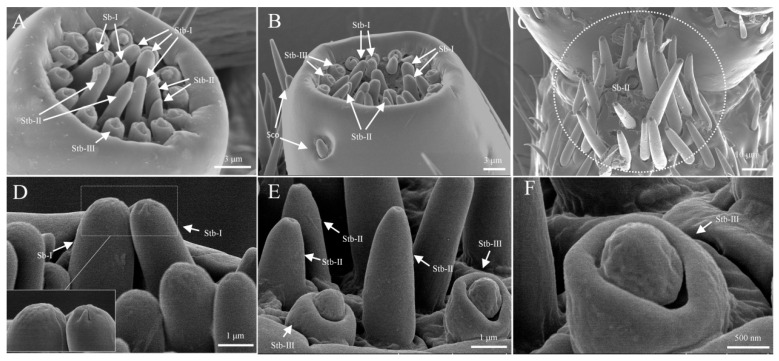
Ultrastructure of sensilla types on the distal regions of the maxillary palps, labial palps, and the ventral surface of the labium. (**A**) Sensory field at the tip of the labial palp; (**B**) sensory field at the tip of the maxillary palp; (**C**) s. basiconica II on the ventral surface of the labium; (**D**) s. basiconica I and s. twig basiconica I at the tips of the maxillary and labial palps; (**E**) s. twig basiconica II at the tips of the maxillary and labial palps; (**F**) s. twig basiconica III at the tips of the labial palps.

**Figure 8 insects-16-00890-f008:**
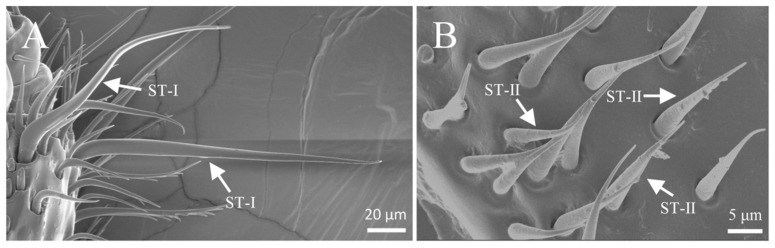
Morphology of sensilla trichoidea on mouthparts. (**A**) S. trichoidea I on mouthparts; (**B**) s. trichoidea II on mouthparts.

**Figure 9 insects-16-00890-f009:**
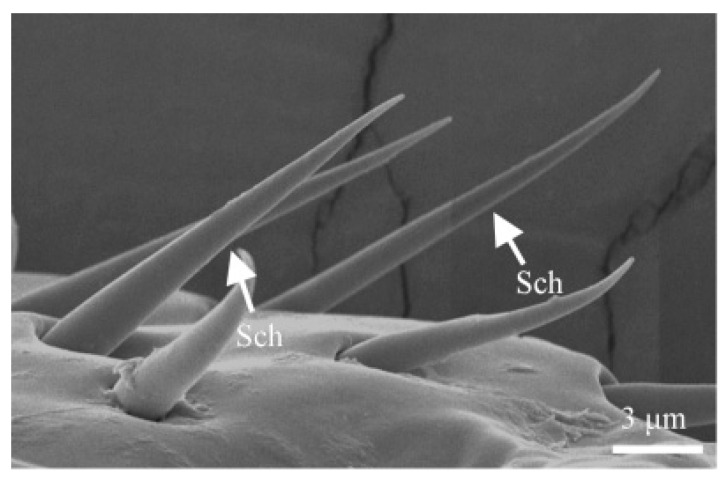
S. chaetica on mouthparts.

**Figure 10 insects-16-00890-f010:**
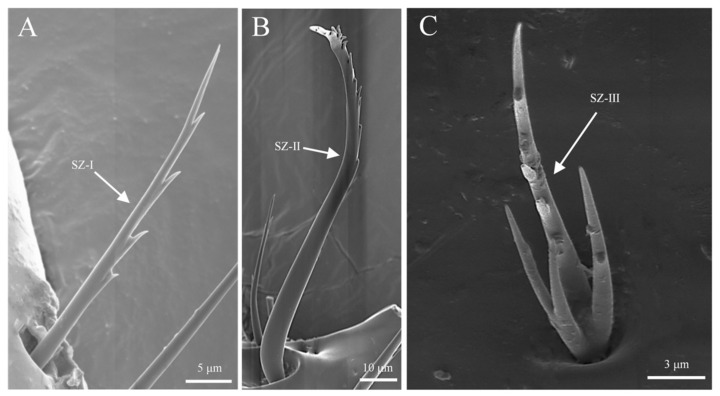
Morphology of sensilla zigzag on mouthparts. (**A**) S. zigzag I on mouthparts; (**B**) s. zigzag II on mouthparts; (**C**) s. zigzag III on mouthparts.

**Figure 11 insects-16-00890-f011:**
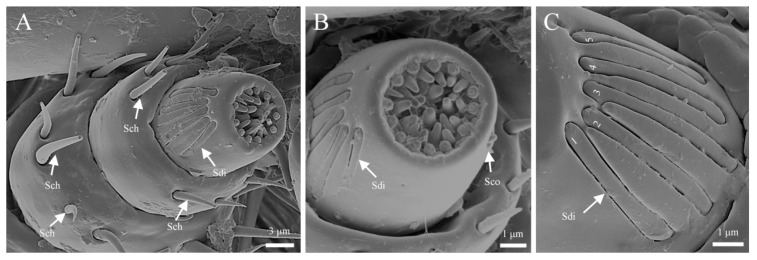
Morphology of the digitiform sensilla on the mouthparts. (**A**) Position of sensilla digitiformia on the maxillary palp. (**B**) Distribution of sensilla digitiformia and sensilla coeloconica on the mouthparts. (**C**) Sensilla digitiformia on the mouthparts.

**Figure 12 insects-16-00890-f012:**
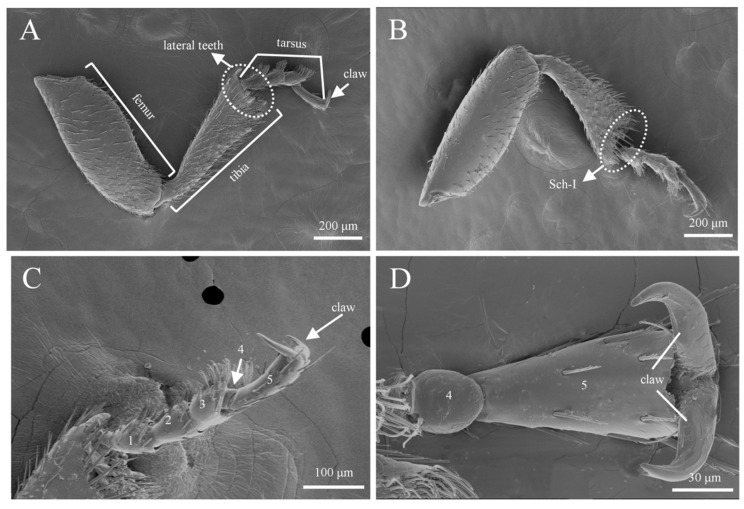
General morphology and structure of the leg of *T. pilifer*. (**A**) The lateral surface of the forelegs; (**B**) the inner surface of the forelegs; (**C**) the structure of the tarsus, 1–5: the 1st to 5th subsegments of the tarsus; (**D**) the shape of the claws.

**Figure 13 insects-16-00890-f013:**
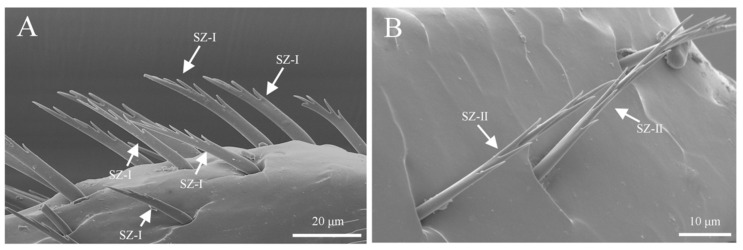
Morphology of sensilla zigzag on the leg. (**A**) S. zigzag I on the leg; (**B**) s. zigzag II on the leg.

**Figure 14 insects-16-00890-f014:**
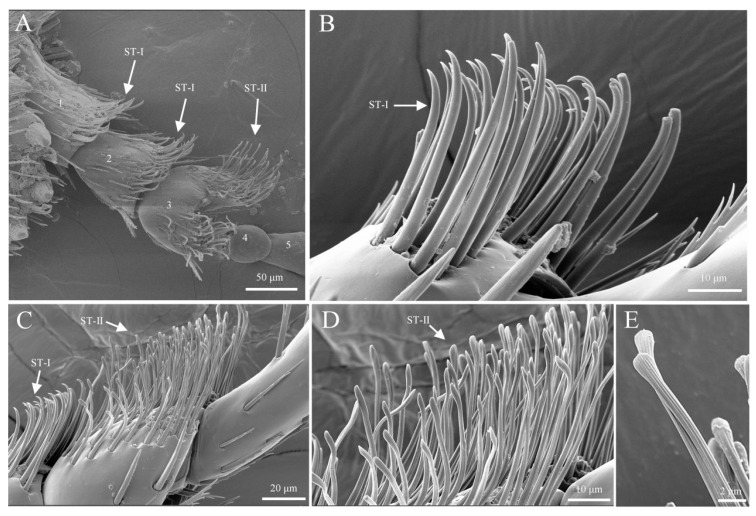
Morphology of sensilla trichoidea on the leg, 1-5: segments 1–5 of the podomere. (**A**) The distribution of s. trichoidea on the tarsus; (**B**) s. trichoidea I on the leg. (**C**) The distribution of s. trichoidea II on the tarsus; (**D**) s. trichoidea II on the leg; (**E**) the surface of s. trichoidea II.

**Figure 15 insects-16-00890-f015:**
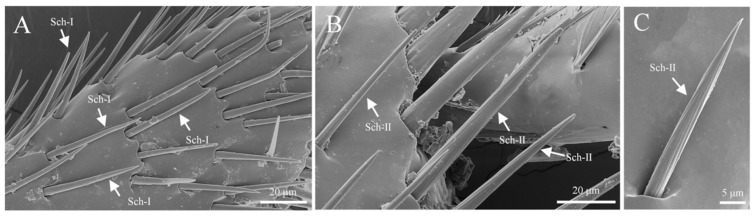
Morphology of sensilla chaetica on the foot. (**A**) S. chaetica I on the leg; (**B**) s. chaetica II on the leg; (**C**) the surface of s. chaetica II.

**Table 1 insects-16-00890-t001:** Distribution of sensilla on the antennae of male and female *Tomicus pilifer* adults.

		ST I		ST Ⅱ		SZ I		SZ Ⅱ		Sco		Sch		Bb		Sb I		Sb Ⅱ		Sb Ⅲ		Sb Ⅳ	
		♂	♀	♂	♀	♂	♀	♂	♀	♂	♀	♂	♀	♂	♀	♂	♀	♂	♀	♂	♀	♂	♀
Scape		-	-	-	-	17.3 ± 2.8	18.1 ± 2.3	-	-	-	-	-	-	14.7 ± 0.8	14.3 ± 1.1	-	-	-	-	-	-	-	-
Funicle	F1	-	-	-	-	4.7 ± 0.6	4.3 ± 0.2	1.3 ± 0.2	1.1 ± 0.1	-	-	2.3 ± 0.2	2.7 ± 0.5	-	-	-	-	-	-	-	-	-	-
	F2	-	-	-	-	-	-	-	-	-	-	-	2.3 ± 0.2	-	-	-	-	-	-	-	-	-	-
	F3	-	-	-	-	2.1 ± 0.2	2.3 ± 0.2	1.3 ± 0.1	1.7 ± 0.4	-	-	-	-	-	-	-	-	-	-	-	-	-	-
	F4	-	-	-	-	3.7 ± 0.6	3.9 ± 0.5	-	-	-	-	-	-	-	-	-	-	-	-	-	-	-	-
	F5	-	-	-	-	1.9 ± 0.3	1.7 ± 0.4	-	-	-	-	-	-	-	-	-	-	-	-	-	-	-	-
	F6		-	-	-	7.8 ± 1.6	8.1 ± 1.5	-	-	-	-	-	-	-	-	-	-	-	-	-	-	-	-
Capitate	C1	62.9 ± 11.3	59.7 ± 9.7	-	-	38.7 ± 6.2	41.2 ± 5.7	-	-	8.7 ± 2.1	9.6 ± 2.4	-	-	-	-	127.3 ± 13.9	121.2 ± 11.6	33.7 ± 5.9	36.8 ± 7.4	-	-	-	-
	C2	51.5 ± 9.2	47.8 ± 8.7	-	-	18.5 ± 3.1	16.4 ± 2.7	-	-	13.2 ± 2.5	13.7 ± 2.7	-	-	-	-	117.6 ± 17.3	114.8 ± 15.3	27.1 ± 3.1	31.8 ± 3.5	-	-	-	-
	C3	47.1 ± 6.8	46.7 ± 7.2	3.6 ± 0.7	3.7 ± 0.4	-	-	-	-	9.1 ± 1.5	8.6 ± 1.7	18.2 ± 2.3	17.5 ± 2.7	-	-	41.2 ± 3.7	42.7 ± 3.5	21.2 ± 3.7	22.7 ± 3.5	-	-	-	-
	C4	12.9 ± 2.7	13.7 ± 3.2	6.9 ± 1.5	6.7 ± 1.2	-	-	-	-	-	-	13.5 ± 1.2	13.8 ± 1.7	9.5 ± 0.5	9.2 ± 0.7	14.7 ± 2.9	13.1 ± 2.6	14.7 ± 2.9	13.1 ± 2.6	-	28.6 ± 6.1	8.7 ± 1.2	9.1 ± 1.4

Note: Data are the mean ± S.E.

**Table 2 insects-16-00890-t002:** Length, width, morphological characteristics, and quantity of various types of sensilla in female and male adults of *T. pilifer*.

Sensillum Type	Sex	Length (μm)	Basal Width (μm)	Exine	Tip	Shape	Total
ST Ⅰ	♂	15.6 ± 1.7	1.4 ± 0.3	stripe	sharp	erect	174.4 ± 29.7
	♀	15.7 ± 1.5	1.4 ± 0.2	stripe	sharp	erect	167.9 ± 28.8
ST Ⅱ	♂	34.3 ± 3.2	2.1 ± 0.4	stripe	sharp	bend	10.5 ± 2.2
	♀	34.1 ± 3.1	2.1 ± 0.4	stripe	sharp	bend	10.4 ± 1.6
SZ Ⅰ	♂	23.1 ± 12.7	2.7 ± 0.4	sawtooth	sharp	erect	94.7 ± 15.4
	♀	23.2 ± 12.7	2.6 ± 0.4	sawtooth	sharp	erect	95.9 ± 13.5
SZ Ⅱ	♂	15.3 ± 1.9	1.3 ± 0.4	sawtooth	blunt	erect	3.1 ± 0.4
	♀	15.3 ± 1.8	1.5 ± 0.4	sawtooth	blunt	erect	3.3 ± 0.4
Sco	♂	8.6 ± 0.7	3.7 ± 0.2	stripe	sharp	erect	30.8 ± 6.1
	♀	8.8 ± 0.7	3.5 ± 0.3	stripe	sharp	erect	31.9 ± 6.8
Sch	♂	7.6 ± 0.7	2.9 ± 0.2	stripe	sharp	erect	36.9 ± 8.7
	♀	7.7 ± 0.7	2.8 ± 0.3	stripe	sharp	erect	36.4 ± 8.2
Bb	♂	2.4 ± 0.3	1.1 ± 0.1	smooth	sharp	erect	14.7 ± 0.8
	♀	2.4 ± 0.2	1.3 ± 0.1	smooth	sharp	erect	14.3 ± 0.1
Sb Ⅰ	♂	11.2 ± 0.9	2.1 ± 0.2	pore	blunt	bend	295.3 ± 17.1
	♀	11.7 ± 1.1	2.1 ± 0.3	pore	blunt	bend	293.8 ± 16.1
Sb Ⅱ	♂	11.6 ± 1.3	1.8 ± 0.2	pore	blunt	bend	96.4 ± 19.3
	♀	11.3 ± 1.5	1.9 ± 0.2	pore	blunt	bend	93.8 ± 16.1
Sb Ⅲ	♂	7.7 ± 0.6	3.1 ± 0.2	pore	blunt	bend	28.6 ± 6.1
	♀	-	-	-	-	-	-
Sb Ⅳ	♂	5.7 ± 0.6	1.6 ± 0.2	pore	blunt	bend	8.7 ± 0.6
	♀	5.8 ± 0.8	1.6 ± 0.3	pore	blunt	bend	9.1 ± 1.4

Note: Data are the mean ± S.E.

**Table 3 insects-16-00890-t003:** The number, size, and distribution of various types of sensilla in the mouthparts of female and male adults of *T. pilifer*.

Sensilla	Total		Length (μm)		Basal Width (μm)		Location
	♂	♀	♂	♀	♂	♀	
Sb Ⅰ	13.2 ± 0.6	13.7 ± 0.6	5.2 ± 1.2	5.1 ± 0.9	0.8 ± 0.2	0.8 ± 0.3	Mpt, Lpt
Sb Ⅱ	42.7 ± 5.9	43.1 ± 5.7	28.7 ± 5.3	28.4 ± 5.2	5.1 ± 1.2	5.3 ± 1.3	Lig
Stb Ⅰ	14.9 ± 1.7	15.3 ± 1.7	2.4 ± 0.2	2.3 ± 0.2	1.2 ± 0.1	1.1 ± 0.1	Mpt, Lpt
Stb Ⅱ	27.4 ± 3.7	27.1 ± 3.2	3.1 ± 0.2	3.2 ± 0.2	0.9 ± 0.1	0.9 ± 0.1	Mpt, Lpt
Stb Ⅲ	58.7 ± 6.7	58.3 ± 6.3	1.4 ± 0.2	1.4 ± 0.3	1.2 ± 0.1	1.2 ± 0.1	Mpt, Lpt
Sco	5.4 ± 0.3	5.2 ± 0.3	1.7 ± 0.2	1.7 ± 0.3	0.8 ± 0.1	0.8 ± 0.1	Mpt, Lpt
ST Ⅰ	96.7 ± 11.5	101.4 ± 11.7	87.1 ± 21.2	86.6 ± 21.3	5.1 ± 1.2	5.3 ± 1.3	Mx, Lr, Li
ST Ⅱ	137.5 ± 12.9	135.1 ± 12.2	14.2 ± 2.6	13.9 ± 3.1	2.3 ± 0.7	2.3 ± 0.5	Glc
Sch	119.7 ± 8.4	119.2 ± 8.7	14.2 ± 2.7	14.5 ± 2.5	1.7 ± 0.4	1.7 ± 0.4	Md, Mx, Li
SZ Ⅰ	51.1 ± 7.2	48.6 ± 7.8	34.8 ± 2.1	34.2 ± 2.6	1.9 ± 0.3	1.8 ± 0.4	Mx, Li
SZ Ⅱ	18.7 ± 1.7	18.2 ± 1.6	107.3 ± 19.6	105.9 ± 16.2	5.9 ± 0.7	5.7 ± 0.6	Mx
SZ Ⅲ	13.2 ± 2.3	11.6 ± 2.7	7.5 ± 0.7	7.2 ± 0.7	2.8 ± 0.4	2.6 ± 0.5	Li
Sdi	12.7 ± 0.6	13.1 ± 0.7	13.7 ± 0.8	13.2 ± 0.7	1.3 ± 0.2	1.3 ± 0.2	Mpl

Note: Data are the mean ± S.E.

**Table 4 insects-16-00890-t004:** The number of sensilla of different types on the front, middle, and hind legs of female and male adults of *T. pilifer*.

Sensilla	Forelegs		Midlegs		Hindlegs	
	♂	♀	♂	♀	♂	♀
SZ Ⅰ	687.9 ± 57.4	677.5 ± 53.2	685.7 ± 49.6	681.2 ± 51.3	691.2 ± 53.2	682.3 ± 51.7
SZ Ⅱ	43.2 ± 7.6	45.3 ± 7.2	43.9 ± 6.8	43.1 ± 6.9	44.9 ± 7.3	43.7 ± 7.0
ST Ⅰ	132.2 ± 19.6	131.7 ± 19.2	129.7 ± 21.3	131.4 ± 20.7	131.9 ± 19.7	131.5 ± 19.4
ST Ⅱ	87.2 ± 9.7	88.5 ± 9.2	87.6 ± 9.2	87.5 ± 9.5	86.5 ± 10.1	87.3 ± 9.8
Sch Ⅰ	76.5 ± 11.6	79.7 ± 10.2	77.4 ± 10.8	77.6 ± 11.1	78.2 ± 10.9	77.9 ± 10.7
Sch Ⅱ	31.4 ± 6.7	30.8 ± 6.3	31.7 ± 6.5	32.4 ± 6.9	32.7 ± 6.2	31.8 ± 6.7

Note: Data are the mean ± S.E.

## Data Availability

We agree to share existing datasets or raw data that have been analyzed in the manuscript, and they will be made available to other researchers following publication. Data availability status: All data are contained within the article and Appendix A.

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
