# Peer review of "Ultrastructural Morphology and Descriptive Analysis of Cuticular Sensilla in Adult Tomicus pilifer (Coleoptera: Curculionidae)"

_insects, 2025, doi:10.3390/insects16090890_

Round 1
Reviewer 1 Report
Comments and Suggestions for Authors
Dear authors,
I find the article interesting and highly important for understanding the ecology of these insects, therefore, I would suggest it for publication. However, please take a look at the comments in the pdf file and make sure that many typos and mistakes are corrected.
Wish you all the best and a fast publication process

Reviewer 2 Report
Comments and Suggestions for Authors
The manuscript presents a very strong descriptive work on the ultrastructure of cuticular sensilla in adult Tomicus pilifer, with high-quality SEM pictures and measures. However, I do not think there is any strong reason for the 28S sequence; it is totally unnecessary detail. As a "major" forestry pest, this species should be easily identified by the authors, so there is no need to confirm it using 28S blast or phylogenetic analyses. Admittedly, it is not a fatal problem.
Three specific recommendations:
- Add picture to show the habitus or living status of the species for the general readers;
- The first paragraph of Discussion looks too long; please split it into two or three paragraphs.
- In the suppl. file Appendix A: Table S1. The "Reference" column confused the reference of the sequence with the reference of the name. Here I suggest they move the authors and year of the species name to the first column, following the corresponding name; the "Reference" column should be left for the references that presented the molecular sequences, if applicable.
I am not the native speaker but feel that the authors may improve the language quality if they want.
